



# J-GAIN v1.0: A flexible tool to incorporate aerosol formation rates obtained by molecular models into large-scale models

Daniel Yazgi[1] and Tinja Olenius[1]

[1]Swedish Meteorological and Hydrological Institute (SMHI), SE-601 76 Norrköping, Sweden

**Correspondence:** Daniel Yazgi (daniel.yazgi@smhi.se), Tinja Olenius (tinja.olenius@alumni.helsinki.fi)

**Abstract.** New-particle formation from condensable vapors is a common atmospheric process that has significant but uncertain effects on aerosol particle number concentrations and impacts. Assessing the formation rates of nanometer-sized particles from different vapors is an active field of research within atmospheric sciences, with new data being constantly produced by molecular models and experimental studies. Such data can be implemented in large-scale climate and air quality models

as parameterizations or look-up tables. Models benchmarked against measurement data provide a straight-forward means to assess formation rates over a wide range of atmospheric conditions for given chemical compounds. Ideally, the implementation of such formation rate data should be easy, efficient and flexible in the sense that same tools can be conveniently applied for different data sets in which the formation rate depends on different parameters. In this work, we present a tool to generate and interpolate look-up tables of formation rates for user-defined input parameters. The table generator routine applies a molecular

cluster dynamics model with quantum chemistry input, but other types of particle formation models may be used as well. The interpolation routine uses a multivariate interpolation algorithm, which is applicable to different numbers of independent parameters, and gives fast and accurate results with typical interpolation errors of up to a few percent. These routines facilitate the implementation and testing of different aerosol formation rate predictions in large-scale models, allowing straight-forward inclusion of new or updated data without the need to apply separate parameterizations or routines for different data sets that

involve different chemical compounds or other parameters.

## 1 Introduction

Formation of secondary aerosol particles from condensable vapors is a well-known and ubiquitous phenomenon in Earth's atmosphere (Kerminen et al., 2018; Kontkanen et al., 2017). Aerosols have significant effects on climate and air quality, with the indirect climate effect through aerosol–cloud interactions forming the largest uncertainty in radiative forcing assessments

(Naik et al., 2021). New-particle formation, including the formation and condensational growth of nanoparticles from vapors, is an important factor affecting aerosol number concentrations and size distributions (Fountoukis et al., 2012; Makkonen et al., 2012; Gordon et al., 2017). The primary quantity characterizing the formation process is the initial particle formation rate $J$, often referred to as the nucleation rate, which gives the rate at which new particles of ca. 1–2 nm in diameter form per unit time and volume at given ambient conditions through clustering of vapor molecules. The chemical compounds that are able to drive

and enhance the initial formation process include acids, bases, and organic species (Glasoe et al., 2015; Lehtipalo et al., 2018;



Xiao et al., 2021). The roles and effects of different potentially important precursor vapors are not, however, resolved. While it is established that sulfuric acid ($H_2SO_4$) initiates particle formation in the presence of stabilizing species such as ammonia ($NH_3$) or monoamines (Jen et al., 2014; Almeida et al., 2013), there exist various species, such as diamines (Jen et al., 2016), organic acids (Zhang et al., 2004) and complex highly oxidized organic molecules (Kirkby et al., 2016), that may contribute

to particle formation with or without sulfuric acid. Furthermore, quantitative particle formation rates are very challenging to assess: both theoretically and experimentally deduced formation rates involve high uncertainties (Almeida et al., 2013; Kürten et al., 2018), and may be sensitive to ambient conditions such as temperature and scavenging sink of the initial molecular clusters (Olenius et al., 2017; Elm et al., 2020).

The combination of quantum chemistry and molecular cluster dynamics simulations is the state-of-the-art method to cal-

culate theoretical particle formation rates (Elm et al., 2020). Quantum chemistry is applied to compute the thermodynamics of the formation of the initial molecular clusters from vapor molecules. The thermochemistry data is used as input in cluster population dynamics simulations, which determine the formation rate by modeling cluster formation and growth considering molecular collisions and evaporations and other dynamic processes such as cluster sinks (Olenius et al., 2013). Such simulations are fast and easy to perform, thus enabling the assessment of formation rates over wide ranges of atmospheric conditions

for given chemical compounds. Benchmarking the theoretical methods against experimental data, which often cover a limited set of conditions, provides quality control and uncertainty estimates for the predicted formation rates (Almeida et al., 2013; Kürten et al., 2016).

New quantum chemical data, calculated for different chemical species (Elm et al., 2020; Elm, 2019a) or by applying different physical and computational approaches (Besel et al., 2020), is constantly becoming available. Such data can be used to test the

effects of different particle formation mechanisms or improved quantitative formation rate predictions in atmospheric large-scale models. To this end, the formation rate data must be implemented as parameterizations or look-up tables (Dunne et al., 2016; Ehrhart et al., 2018; Kazil et al., 2010; Yu et al., 2020; Baranizadeh et al., 2016), as applying formation rate simulations within a large-scale model is not feasible. The formation rate is a function of several parameters, the most obvious of which are the concentrations of the vapors participating in the formation process, and the temperature. In addition, formation rates

obtained by cluster simulations include the effects of cluster sink caused by pre-existing larger aerosol particles and, depending on the chemical compounds and quantum chemical input data, possibly also atmospheric ions and relative humidity. In the presence of several cluster-forming vapors, particle formation can proceed through multicompound clustering, or parallel clustering pathways involving non-interacting chemical mechanisms (Elm, 2019b).

Deriving parameterizations quickly becomes cumbersome as the number of independent parameters increases: finding pa-

rameterization formulas and coefficients that reproduce the formation rate data with a reasonable accuracy throughout the parameter space is virtually impossible for arbitrary chemical systems. Furthermore, evaluation of very complex formulas within a computationally heavy large-scale model may not be optimal. The benefit of look-up tables is that values determined from a table of sufficient resolution are guaranteed to be close the original data, and no pre-processing of the formation rate data is needed. The use of such tables, on the other hand, requires multivariate interpolation algorithms that should ideally be

applicable to tables of arbitrary dimensions, with no need for manual changes depending on the number of independent param-



eters. While interpolated values must be sufficiently accurate, the interpolation routine should be computationally efficient and preferably simple.

In this work, we introduce a tool to incorporate molecular modeling data in atmospheric models by flexible routines to generate and interpolate formation rate look-up tables. With this approach, we aim to cover the following aspects:

– flexible implementation of state-of-the-art molecular modeling results in large-scale models,

   – inclusion of arbitrary chemical compounds,

   – efficient multivariate interpolation, and

   – user-friendly routines with no need for modifications depending on, for instance, included chemical compounds.

We apply a publicly available cluster dynamics model to generate formation rate look-up tables from user-defined quantum
chemistry input data, and provide a table interpolation routine that can be readily implemented in a large-scale model. We present assessments of optimal table resolution and interpolation accuracy, and demonstrate the use and performance of the tool by application on sulfuric acid—ammonia particle formation rate data. We generate tables suitable for global applications for these data, and assess the performance of the interpolation routine in typical computationally heavy model set-ups.

## 2   Methods

The proposed look-up table approach comprises two components: a generator routine to create tables, and an interpolation routine to be implemented in an atmospheric model. We refer to the routines as J-GAIN (Formation rate ($J$) look-up table Generator And INterpolator). The details of the generation and interpolation routines are presented in Sects. 2.1 and 2.2, respectively, and their application is further discussed in Sects. 2.3 and 2.4. Briefly, the table generation is designed for input data from quantum chemical calculations, which is the standard method in theoretical atmospheric particle formation studies.
In principle, also formation rate data obtained by other means can be used, given that the table format is similar and thus compatible with the interpolation routine. Fig. 1 shows a schematic presentation of the J-GAIN routines, summarizing the input, output and main features. J-GAIN is written in Fortran, with the exception that the molecular cluster dynamics model used by the table generator routine applies also Perl, and licensed under GPL 3.0. The code repository includes detailed instructions for using the programs (Yazgi and Olenius, 2022a, b).

### 2.1   Table generation

The table generator routine requires the following input: molecular cluster thermochemistry data for the given chemical compounds, and user-defined input for the ranges of the parameters that define the ambient conditions. These parameters include the concentrations of the vapor compounds, temperature ($T$), coagulational scavenging sink (CS; given as vapor condensation sink which is scaled for different cluster sizes within the cluster dynamics model as derived by Lehtinen et al. (2007)), and
optionally also ion production rate (IPR; given as generic ion pairs per unit volume and time) and relative humidity (RH).



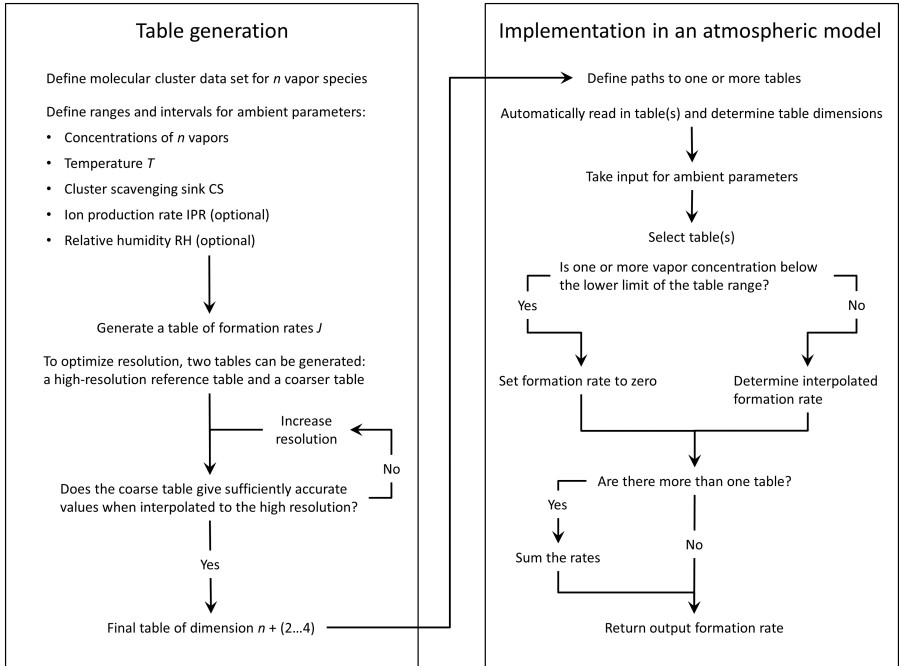

**Figure 1.** Flow chart illustrating the generation and application of particle formation rate tables by J-GAIN.

This gives $n + (2 \ldots 4)$ independent parameters, where $n$ is the number of chemical compounds. While vapor concentrations, temperature and cluster sink always affect the particle formation rate and are thus included by default, the inclusion of IPR and RH depends on the chemical system, and also on data availability. For strongly clustering chemistries, ion effects may be negligible (Myllys et al., 2019), and also RH effects may be minor (Henschel et al., 2016). As the inclusion of charged species

and hydrates in a molecular cluster data set requires a significant computational effort, these effects are not always available.

The user sets the range for the values of each parameter by giving the lower and upper limits, and the number of values to be placed within the limits at even intervals. Parameters can be defined as "logarithmic", in which case the data points are placed evenly on a logarithmic scale. This is relevant for, for example, vapor concentrations. The formation rate table is then generated by calling the molecular cluster dynamics model ACDC to obtain the formation rate $J$ for each combination of

parameter values. The details of the ACDC model can be found in the work by Olenius et al. (2013) and in the ACDC code repository (Olenius, 2021).

The tables are outputted as binary files, and a descriptor file is generated together with the table. The latter contains the essential information on the table, including the names and units of the independent parameters, the lower and upper limits of the parameter values, and the numbers of values along each dimension. In order to ensure a sufficient accuracy, several

tables can be generated at different resolutions. Then, possible errors in interpolated values can be assessed by interpolating a coarser-resolution table on the grid of a higher-resolution reference table (Sect. 3.1).



## 2.2 Table interpolation

The interpolation routine uses the descriptor file to obtain the number and identities of the independent parameters for the corresponding look-up table. After loading the table, the routine determines the formation rate by linear multivariate interpolation. In general, for an $N$-dimensional function $f = f(x_1, x_2, ..., x_N)$ with nearest known data points at locations $x_{j,0}$ and $x_{j,1}$ for each variable $x_j$, where $x_{j,0} < x_{j,1}$ and $j = 1, \ldots, N$, the interpolated value $\tilde{f}$ can be written as

$$\tilde{f}(x_1, x_2, ..., x_N) = \sum_{(i_1, i_2, ..., i_N) \in \{0,1\}^N} f(x_{1,i_1}, x_{2,i_2}, ..., x_{N,i_N}) \prod_{j=1}^{N} \begin{cases} x'_j, & i_j = 1 \\ 1 - x'_j, & i_j = 0 \end{cases}, \tag{1}$$

where the summation goes over the values of $f$ at all combinations of points $x_{j,0}$ and $x_{j,1}$ for $j = 1, \ldots, N$, and

$$x'_j = \frac{x_j - x_{j,0}}{x_{j,1} - x_{j,0}}. \tag{2}$$

Here, we perform linear interpolation by default for $f = \log J$ and either $x_j$ or $\log x_j$ depending on if parameter $j$ is defined as linear or logarithmic, respectively. This gives more accurate results than assuming purely linear relationships, as $J$ generally varies smoothly on a logarithmic scale. Moreover, a linear dependence between the logarithms of $J$ and the concentration $C$ of a contributing vapor within a short interval along the $C$-axis is expected also based on simplified cluster formation theories (see e.g. Li and Signorell, 2021). The choices of applying either the actual values or the logarithms of the dependent and independent parameters can also be changed by the user.

Treatment of input values that are outside the ranges covered by the table is parameter-dependent and defined by the user. For each parameter, the following options are included in the example routines:

1. Value below/above the table limits → the value is set to the lower/upper limit of the table (this is the default behavior)

2. Value below the lower limit → $J = 0$ is returned; input value above the upper limit → the value is set to the upper limit

The latter option is for vapor concentrations, provided that the lower limits of the table are chosen so that effectively no particle formation is expected at concentrations below the limits. The first option is by default used for other parameters.

## 2.3 Incorporation of tables in a host model

In the code repository, we provide an example of simple interfaces to load and interpolate look-up tables within a host model. The input parameters of the interpolation subroutine include all parameters that the particle formation rate may depend on, and the total formation rate is returned as output. Separate particle formation pathways, corresponding to different chemical compounds, can be incorporated as separate tables. If more than one table are defined in the subroutine, the total formation rate is obtained as the sum of the individual formation rates.



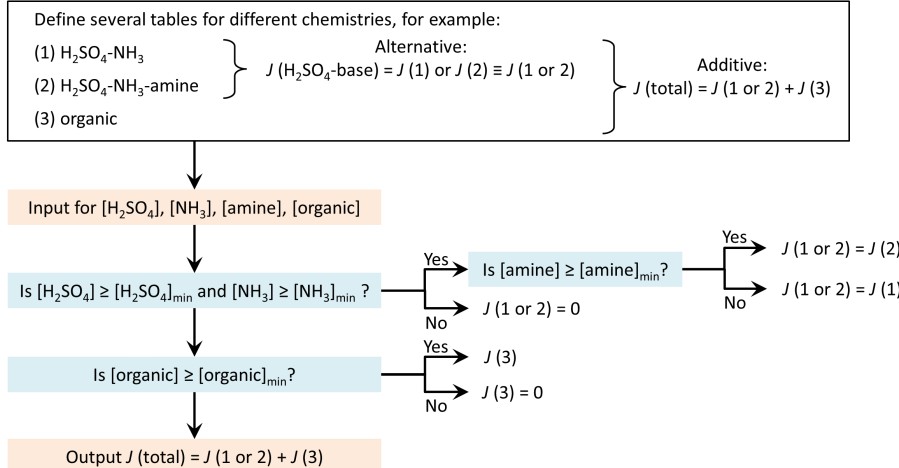

**Figure 2.** Schematic presentation of the treatment of several tables.

The user may construct different ways to treat several tables according to their needs. There may also be alternative tables depending on the presence of a given chemical species, that is, if the concentration of the species is above the threshold given by the lower limit in the relevant table. For instance, there may be data for particle formation from $H_2SO_4$ and $NH_3$ with or without an amine species. In the presence of the amine, a table of $H_2SO_4$—$NH_3$–amine formation rates is selected, while otherwise a $H_2SO_4$—$NH_3$ table is applied. The selection and summation of tables is schematically shown in Fig. 2.

## 2.4 Example case: sulfuric acid—ammonia particle formation

We demonstrate the application and performance of J-GAIN using previously published quantum chemical data for sulfuric acid and ammonia (Olenius et al., 2013), which are common atmospheric particle formation precursor vapors. Here, the input data for the $H_2SO_4$-–$NH_3$ system includes charged clusters but no hydration, and therefore the independent parameters that determine the formation rate are vapor concentrations $[H_2SO_4]$ and $[NH_3]$, $T$, CS and IPR.

We apply the $H_2SO_4$—$NH_3$ data to generate tables of different resolution, and compare the interpolated values of $J$ against accurate values given by the highest-resolution reference table. We also generate tables suitable for global applications, covering ranges relevant for boundary layer environments and upper troposphere for all independent parameters. These extended tables are evaluated by determining the accuracy of $J$ for conditions where all parameters $[H_2SO_4]$, $[NH_3]$, $T$, CS and IPR vary with time, corresponding to practical model implementations. To assess the performance of the interpolation routine for given table sizes, we determine the interpolation speed for test tables of different numbers of data points and dimensions.





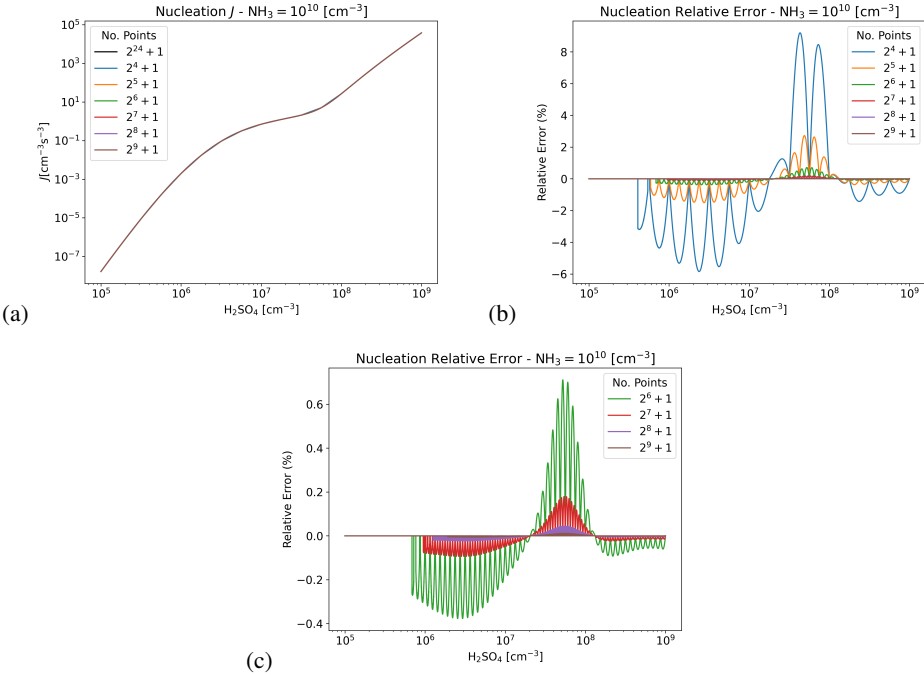

**Figure 3.** Particle formation rate $J$ as a function of $H_2SO_4$ concentration for $H_2SO_4$—$NH_3$ model chemistry ($[NH_3] = 10^{10}$ cm$^{-3}$, $T = 298.15$ K, CS $= 10^{-3}$ s$^{-1}$, IPR $= 3$ cm$^{-3}$s$^{-1}$), determined from look-up tables of different resolution. (a) Absolute $J$. (b) Relative error in interpolated $J$ compared to accurate values given by the highest-resolution reference table with $2^{24} + 1$ points along the $[H_2SO_4]$-axis. (c) Relative error in interpolated $J$ for resolutions of $\geq 2^6 + 1$ points.

## 3 Results and discussion

### 3.1 Effect of table resolution on accuracy

Fig. 3 presents $J$ interpolated along the $[H_2SO_4]$-axis and the error in the interpolated values at representative atmospheric conditions for tables with $2^k + 1, k = 4...9$ data points on the axis. For the lowest resolution of $2^4 + 1 = 17$ points, the relative error is less than $\pm 10$ %, and for $\geq 2^6 + 1 = 65$ points the error is well below $\pm 1$ %. Fig. 4 shows the error as a function of both $[H_2SO_4]$ and $[NH_3]$ at two different temperatures for the two lowest resolutions used here. For $2^4 + 1 = 17$ points along both axes, the error is up to 10 % but mostly below it, and doubling the resolution to $2^5 + 1 = 33$ points drops the error below a couple of percent.

In order to demonstrate the performance of the interpolation approach at realistic ambient conditions at which all parameters can vary with time, corresponding to implementation of the routine in an atmospheric model, $J$ is determined for a representative diurnal cycle as shown in Fig. 5. The independent parameters are set to follow 24-hour time profiles as described in Appendix A. Here, we apply extended tables that cover wide ranges of parameter values, suitable for larger-scale chemical transport or general circulation models: $[H_2SO_4] = 10^5 - 10^8$ cm$^{-3}$, $[NH_3] = 10^6 - 3 \times 10^{11}$ cm$^{-3}$, $T = 180 - 320$ K,



(a) $T = 298.15$ K

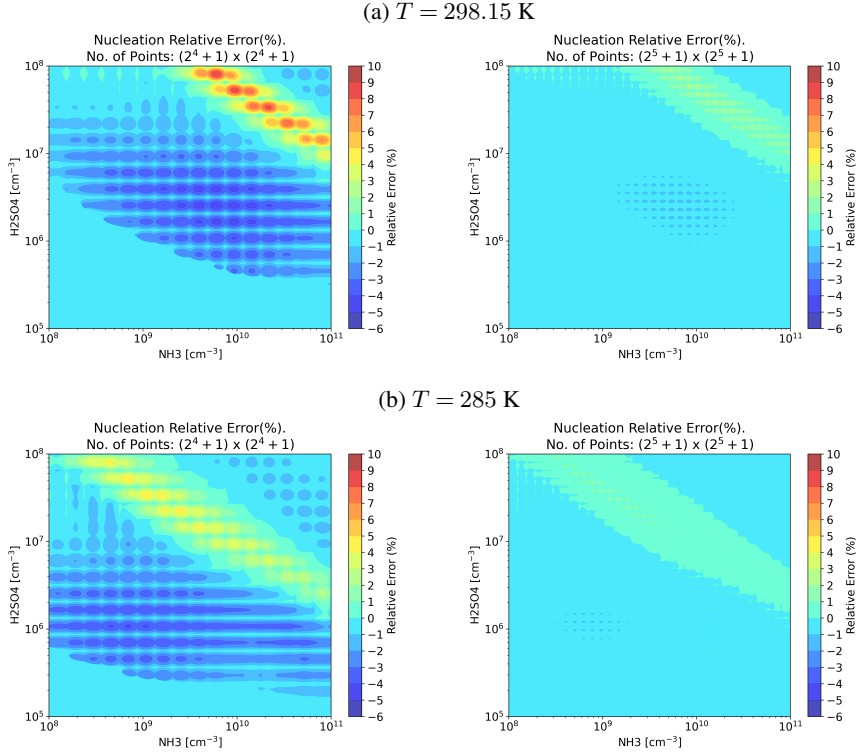

**Figure 4.** Relative error in interpolated particle formation rate $J$ as a function of $H_2SO_4$ and $NH_3$ concentrations for look-up tables with $2^4 + 1$ and $2^5 + 1$ data points along the $[H_2SO_4]$- and $[NH_3]$-axes at $CS = 10^{-3}$ s$^{-1}$, IPR $= 3$ cm$^{-3}$s$^{-1}$, and (a) $T = 298.15$ K, (b) $T = 285$ K.

$CS = 10^{-5} - 10^{-1}$ s$^{-1}$, and IPR $= 0.1 - 60$ cm$^{-3}$ s$^{-1}$. The errors in Fig. 5 are generally of the same order as in Figs. 3 and 4, although somewhat higher due to interpolation over all parameters. Doubling the table resolution from $2^4 + 1 = 17$ to $2^5 + 1 = 33$ points (1) only for vapor concentrations, and (2) for all parameters decreases the maximum errors down to ca.
$\pm 15$ % and well below $\pm 10$ %, respectively. The errors are similar when using different absolute values of the independent parameters, tested by modifying the diurnal profiles by scaling the parameter values up or down (by e.g. an order of magnitude for vapor concentrations or CS).

In general, such errors are very small compared to typical uncertainty estimates for formation rate data: uncertainties in both theoretical and experimental $J$ generally span up to an order of magnitude and beyond (see e.g. Kürten et al., 2016). Based
on the present evaluations, the interpolation approach is not expected to add notable uncertainties in the formation rate representation even at the lowest resolutions applied. Moreover, the accuracy is significantly increased by doubling the resolution. However, it can be noted that high-resolution tables quickly become very large, and the interpolation speed may be affected by the number of data points along the dimensions. Therefore, the choice of resolution depends on (1) the desired accuracy, (2)





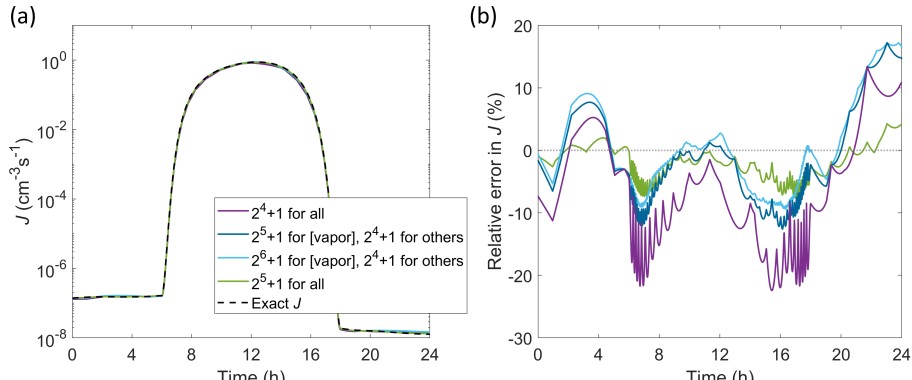

**Figure 5.** Particle formation rate $J$ for a diurnal cycle in which all independent parameters [$H_2SO_4$], [$NH_3$], $T$, CS and IPR vary with time (Appendix A). (a) Absolute $J$. (b) Relative error in interpolated $J$ compared to accurate values. Interpolated values are determined from the extended tables with increasing resolution either along only vapor concentration axes, or along all axes as indicated in the legend.

how large tables the user is willing to generate and store, and (3) the computational aspects of the model application considering
the number of calls to the interpolator (Sect. 3.2).

### 3.2   Effect of table size and number of independent parameters on performance

In order to assess the interpolation speed with respect to the numbers of data points and independent parameters, we apply arbitrary test tables of different sizes and dimensions. Fig. 6 shows the mean run time for one million calls to the interpolator with randomly assigned input parameter values. Here, the $x$-axis is the base-2 logarithm of the table size (total number of
values in the table). Therefore, for a table with equal numbers of points $2^k + 1$ along all dimensions, the $x$-axis corresponds to approximately $N \times k$, where $N$ is the number of dimensions.

The performance is primarily affected by the number of dimensions: the addition of a new dimension may increase the run time by up to a factor of $\sim 2$. In addition, the run time exhibits a major increase when the table size increases beyond ca. $2^{28}$ (of the order of $\gtrsim 10^8$) values per table. For the $H_2SO_4$–$NH_3$ example table with $N = 5$, this corresponds to an increase in run
time at $k > 5$. The absolute run times of Fig. 6 can be compared to typical overall times for e.g. global-scale applications. We have implemented the $H_2SO_4$–$NH_3$ table for first tests in the EC-Earth climate model (Döscher et al., 2022). In this EC-Earth set-up, the total number of grid points where the interpolation routine is called is 367200 ($120 \times 90$ horizontal grid points and 34 vertical layers), the time step for determining the formation rate is one hour, and the chemistry model uses 90 processes. This corresponds to approximately $\lesssim 1$ minute run time for the formation rate routine for a simulation of one year, assuming a
mean time of $\lesssim 2$ seconds for $10^6$ look-ups (Fig. 6, resolutions of $2^4 + 1$ and $2^5 + 1$ for all axes). Compared to the total model run time of $\gtrsim 10$ hours per year, the contribution of the formation rate routine is negligible.

For applications with e.g. higher spatial resolution or shorter time step, the performance of the formation rate routine can be optimized by reasonable choices of table dimensions and size. For example, for very strongly clustering chemical systems,



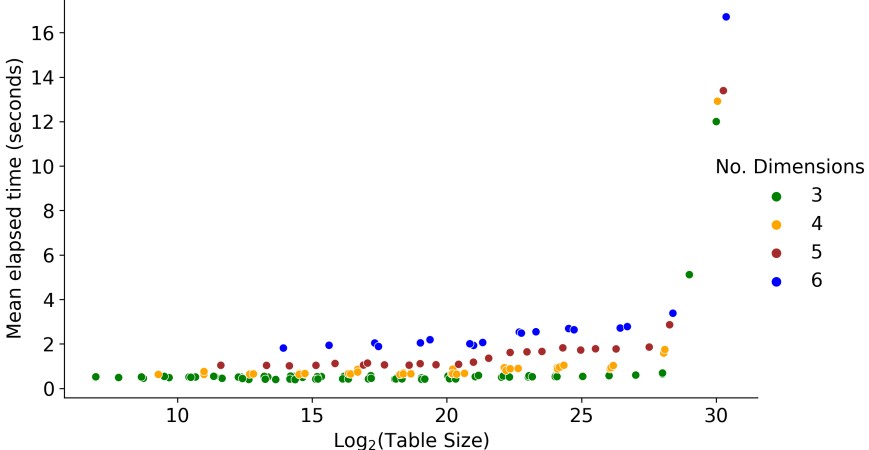

**Figure 6.** Time required to perform one million look-ups for tables of different sizes. The figure shows different combinations $(2^k + 1, 2^l + 1, 2^m + 1, ..)$ of data points along axes, keeping $k = l$ and the total number of points $< (2^7 + 1)^2 (2^4 + 1)^4 \sim 2^{30.37}$. For more details, see discussion in Sect. 3.2.

the presence of atmospheric ions has only minor effects. The IPR parameter could thus be discarded in the interest of speed. In addition to the resolution of the independent parameter values (i.e. the absolute intervals), also their ranges (minimum and maximum values) can be chosen considering the modeled environment, so that redundant values are avoided.

It can also be noted that considering the high accuracy of the interpolated values, interpolation from pre-generated tables is superior compared to the time required to calculate the formation rates by the molecular cluster model. For example, generating the table of $(2^4 + 1)^5 = 1419857$ data points takes 27 hours on a single process, corresponding to a mean time of ca. 0.068 seconds per value. Such times would be infeasible for operational models.

### 3.3 Potential limitations in applying formation rates in a host model

It must be noted that an obvious limitation in applying formation rate data in an atmospheric model is the availability of tracers in the host model. While various chemical compounds may contribute to atmospheric particle formation, including large numbers of new species in a chemical transport model may be cumbersome. In addition, sources of individual chemical species, such as different types of amines or organic acids, may not be well quantified.

Here, we apply particle formation rates from $H_2SO_4$ and $NH_3$ which are common species included in atmospheric models, and comprise a central particle formation pathway according to current understanding (Dunne et al., 2016; Gordon et al., 2017). Tables generated in this work can thus be easily implemented in standard models that include aerosol formation processes. On the other hand, $H_2SO_4$–amine particle formation, for example, requires the inclusion of amine species which are not common tracers despite their potential importance to aerosol formation (Bergman et al., 2015). Different types of amines may also exhibit different particle formation efficiencies, and would thus need to be modeled as separate compounds instead of a lumped



species (Jen et al., 2014; Olenius et al., 2017). Therefore, in order to test the importance of such particle formation agents in a large-scale model, some simplifications are needed.

A simplified option to include rough source–sink dynamics for species that can be assumed to have common sources and
similar properties with respect to gas-phase chemistry and gas-to-particle partitioning is to implement a lumped or representative trace compound. For example, methylamines have been approximated as a representative compound, the emissions of which are estimated based on ammonia emissions (Bergman et al., 2015). To further take into account possible differences in the particle formation rates due to different amines, an option is to estimate the contributions of individual species to the lumped concentration based on available measurements (Schade and Crutzen, 1995). Oxidized organic species can be treated
in a similar manner through a representative highly oxidized, ultra/extremely low-volatile compound (generally referred to as HOM or ULVOC/ELVOC; Kirkby et al., 2016; Schervish and Donahue, 2021), which is already included in some transport models (Gordon et al., 2017; Julin et al., 2018; Patoulias and Pandis, 2022).

In addition, it can be noted that the standard approach to implement formation rates involves certain approximations related to molecular cluster kinetics. Namely, formation rates are assumed to be determined solely by ambient conditions, applying the
steady-state approximation for the cluster population and omitting time-dependent vapor–cluster–aerosol kinetics (Yu, 2003; Olenius and Riipinen, 2017; Olenius and Roldin, 2022). While such simplifications are needed for large-scale applications, they may affect the secondary particle concentrations. To minimize such effects, it can be recommended to extend the modeled aerosol size distribution to as small sizes as achievable instead of extrapolating the formation rate to a substantially larger size (Lee et al., 2013), or to apply a more detailed size distribution representation on the smallest particle sizes (Blichner
et al., 2021). Nevertheless, the inherent steady-state assumption in the initial formation rate at ca. 1 nm is likely to cause overprediction in the case of strongly clustering, low-concentration trace species, and thus the rate should be considered as an upper-limit estimate for such chemistries (Olenius and Roldin, 2022).

## 4   Conclusions

Adequate representation of aerosol particle formation from vapors in atmospheric models is needed for assessing aerosol
climate and health effects. The increasing amount of computational chemistry data enable the assessment of new-particle formation rates for different data sets and chemical compounds by established molecular modeling approaches. As formation rates are typically complicated functions of ambient conditions, they are conveniently incorporated as look-up tables. Here, we provide a tool to generate and interpolate formation rate tables, applicable to arbitrary sets of chemical species, in order to conveniently implement theoretical particle formation rate data in atmospheric large-scale models.

Tests conducted applying theoretical data for $H_2SO_4$-–$NH_3$ particle formation show that the interpolation approach is efficient and accurate, with maximum interpolation errors typically ranging from negligible to ca. $\pm(10 - 20)$ % depending on table resolution. Interpolation errors can also be minimized by choosing to interpolate given parameters on a logarithmic instead of a linear scale. As interpolation speed is affected by the number of independent parameters and also by the table size for large tables (of $\gtrsim 10^6$ values), the choice of parameters and the resolution should be optimized considering the computational



burden of the application where the table is implemented. For heavy applications, redundant independent parameters that have only minor effects on the formation rate can be discarded, and the number of data points along each dimension can be set according to the desired accuracy. The flexibility of the provided tool makes the routines easy to apply to different data sets and choices of table dimensions and size. This design facilitates data transfer between the molecular modeling and the large-scale modeling communities.

## Appendix A: Parameter descriptions in the diurnal test case


Table A1 lists the functional forms of the independent parameters applied for the diurnal test case (Fig. 5). The time profiles, visualized in Fig. A1, are simply set to exhibit realistic magnitudes and shapes: the temperature peaks at daytime while the condensation sink drops as boundary layer grows, and $[\mathrm{H_2SO_4}]$ follows a sinusoidal diurnal profile due to the photochemical production of $\mathrm{H_2SO_4}$. $[\mathrm{NH_3}]$ and ion production rate generally do not show regular diurnal patterns, but are here set to vary in 255 order to test the interpolation approach at different combinations of independent parameters.

**Table A1.** Parameter value as a function of time $t$ (in hours) for the diurnal test case.

| Parameter | Value $f = f(t)$, $t = 0...24$ h |
|---|---|
| $[\mathrm{H_2SO_4}]$ [cm$^{-3}$] | $\begin{cases} 0.5 \times (10^7 - 10^5) \times (\sin(2\pi(t-6)/12 - 0.5\pi) + 1) + 10^5, & t \in \{6...18\} \\ 10^5, & \text{otherwise} \end{cases}$ |
| $[\mathrm{NH_3}]$ [cm$^{-3}$] | $(10^9 - 2 \cdot 10^{10}) \times \exp(-((t-18)/12)^2) + 2 \cdot 10^{10}$ |
| $T$ [K] | $10 \times \exp(-((t-10)/6)^2) + 283.15$ |
| CS [s$^{-1}$] | $(10^{-3} - 6 \cdot 10^{-3}) \times \exp(-((t-10)/12)^2) + 6 \cdot 10^{-3}$ |
| IPR [cm$^{-3}$s$^{-1}$] | $0.5 \times (4 - 1) \times (\sin(2\pi(t-6)/24 - 0.5\pi) + 1) + 1$ |





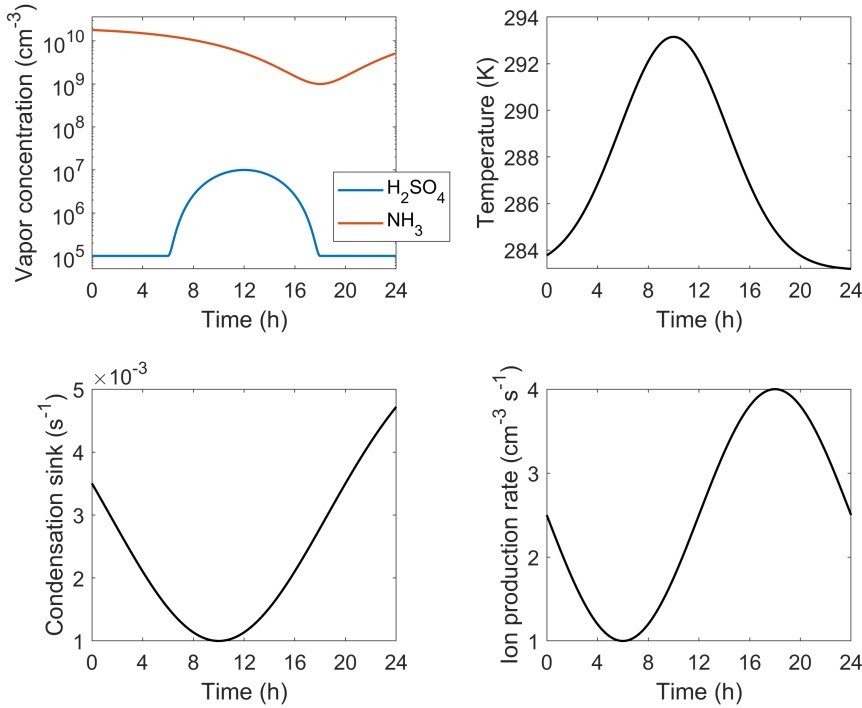

**Figure A1.** Time profiles of the independent parameters for the diurnal test case.

*Code and data availability.* J-GAIN is available at https://github.com/tolenius/J-GAIN (current version) and doi:10.5281/zenodo.7457152 (v1.0). Scripts and tables applied for the figures in this work are available from the corresponding authors.

*Author contributions.* DY designed, wrote and tested the programs, and created the figures. TO conceived the project and wrote the manuscript. All authors contributed to discussing the results and revising the manuscript.

*Competing interests.* The authors declare that they have no conflict of interest.

*Acknowledgements.* The authors gratefully acknowledge financial support from the Swedish Research Council VR (grant no. 2019-04853) and the Swedish Research Council for Sustainable Development FORMAS (grant no. 2019-01433). We thank Pontus Roldin and Carl Svenhag for discussions and help with testing the tool.



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
