# Peer review of "J-GAIN v1.1: A flexible tool to incorporate aerosol formation rates obtained by molecular models into large-scale models"

_EGUsphere, 2022_

## Author Comment (AC1)

We thank both reviewers for their comments that we found very useful for improving our manuscript, and have prepared a revised manuscript accordingly. Please find below our point-by-point replies to all the reviewers' comments. The comments are reproduced in blue and quotes from the revised manuscript are written in red, with the page and line numbers referring to the revised version. We have also revised the whole text and made small modifications to improve readability, and changed figure colors to consider readers with color vision deficiencies. Changes are marked in the manuscript file with yellow highlight.

Reviewer 1

This manuscript presents a tool for efficiently incorporating formation rates from a molecular cluster dynamics model cluster dynamics model with quantum chemistry input into larger models with a look-up table approach. The tool includes a table generator and an interpolator. The table generator calls the molecular cluster dynamics model ACDC for a user-specified ranges of environmental parameters (vapor concentrations, temperature, relative humidity and so on) and generates n-dimensional table data for the particle formation rate. The interpolator can then be included in a host model and reads and interpolates the pre-calculated table data. They present an example case for sulphuric acid-ammonia new particle formation (NPF) and show that the typical errors are negligible to 10-20% for implementation into a global model. These errors seem highly acceptable compared to uncertainties in both theoretical and experimental rates.

The tool seems very useful, and the authors show well how it could be implemented in a global model and therefore seems like a good fit for GMD.

I have two main points I would like the authors to address.

1. In general, it could be made clearer what is done with which parts of the tool.

1. The tool is initially presented as being usable for both measurement data and ACDC data, as well as other models. However, it is not entirely clear upon the first read through whether the table generator is solely used for calling ACDC or whether it could also be used for other sources. On line 80 it is actually explained that it can only be used with ACDC, but because the tool was initially presented in the abstract with "The table generator routine applies a molecular cluster dynamics model with quantum chemistry input, but other types of particle formation models may be used as well." I was still under the impression that the table generator was more generic.

This is a valid comment, and we have now clarified the question of data sources in the revised manuscript (e.g. Introduction; Section 2.1). The table generator is coupled to the ACDC cluster dynamics model because ACDC is a commonly used standard tool for calculation of particle formation rates (as described in Introduction on P2, L35-43). In effect, ACDC is not actually a model but rather an automatic equation generator and solver for the discrete general dynamic equation (DGDE) that describes the molecular cluster kinetics that yield cluster concentrations and formation rates. We feel that it is a practical solution for an automatized table generation procedure because it is applicable to any input molecular cluster set, and can be used without previous experience with input data provided by the quantum chemistry community. Moreover, if an advanced user wishes to apply non-standard or modified model settings, ACDC includes a wide selection of additional features that can be easily used with the help of the comprehensive user manual (for example related to the rate constants of the dynamic processes, such as additional input for collision enhancement factors or sticking probabilities, or user-defined input for non-quantum-chemistry-based evaporation rate constants; ACDC manual, github.com/tolenius/ACDC).

In general, the combination of quantum chemical input data and cluster dynamics simulations has become a standard approach for particle formation modeling (Elm et al., 2020). For this standard method, we encourage to use the table generator with the automatic ACDC solution due to its flexibility, but wish to clarify that using the interpolator is not limited to tables created by a specific program.

The table generator outputs a simple binary table and the descriptor file that summarizes the table contents (as now described in Appendix A and in the 'generator' folder of the J-GAIN repository). However, the J-GAIN interpolator program is independent of the table generation routines, and can be applied to any tables regardless of the table source, as long as the format is correct. This means that formation rate output from other models can, if needed, be saved in a compatible format and used as data source in the interpolator – in fact, it can be noted that the interpolator can in principle be used for any data that lists values of a dependent parameter as a function of multiple independent parameters, i.e. not only formation rate data. We have now included detailed documentation of the table format for such possibility; see the reply to the next comment 1.2 below.

We have made the following modification in the Introduction: the text "We apply a publicly available cluster dynamics model to generate formation rate look-up tables from user-defined quantum chemistry input data, and provide a table interpolation routine that can be readily implemented in a large-scale model" has been replaced with:

We construct a look-up table generator that embeds a publicly available cluster dynamics model to calculate formation rates from user-defined quantum chemistry input data, and provide a table interpolation routine that can be readily implemented in a large-scale model. The table generator enables easy application of the standard formation rate modeling approach, but the interpolator can be used also for tables saved from other models in the same, simple format.

The following text has been added to Section 2.1:

By default, the table generator takes as input molecular cluster thermochemistry data for the given chemical compounds, and calculates the formation rates by molecular cluster dynamics simulations through coupling to the open-source cluster model ACDC (Olenius, 2021). In practice, application of the generator on a new cluster data set consists of two steps: creating the formation rate equations by ACDC, and running the generator to obtain the table (see Yazgi and Olenius, 2022a). As the embedded ACDC application provides automatized and flexible treatment of arbitrary cluster data sets, we recommend the default table generator when applying the commonly used combination of quantum chemical input and cluster dynamics modeling to obtain formation rates. It can be noted that ACDC also includes a wide selection of options that increase the flexibility: for example, while cluster evaporation rate constants are by default obtained from the thermochemistry data, they can also be given as direct input if the user wishes to assess them in some other way. The advanced user may also modify other cluster simulation settings (see the ACDC manual; Olenius, 2021). However, the table interpolator is not limited to tables generated by the default table generator, and thus it is possible to use alternative approaches to determine formation rates. For this, the rates must be saved in the same file format as that produced by the default generator, as detailed in Appendix A.

2. In my opinion, it should be clearer how other formation rate sources could be used with the tool and what pre-processing would need to be done if so. Alternatively, suggestions of such uses could be removed from the abstract (line 9-10).

We have now added information on the table format and instructions for pre-processing data from other sources in Appendix A and the GitHub repository. Appendix A includes a general description, and the 'generator' subdirectory on GitHub summarizes the technical details. We

also include simple example codes for formatting and saving tables in Fortran and C++ in a new folder 'examples/arbitrary_table'.

The sentence in the abstract has been reformulated to read:

The table generator primarily applies cluster dynamics modeling to calculate formation rates from an input quantum chemistry data set defined by the user, but the interpolator may be used also for tables generated by other models or data sources.

3. Related to this, it was unclear to me if the example presented in the result section for H2SO4–NH3 system is made using both the table generator and the interpolator, or if you simply interpolate a pre-calculated table from the cited Olenius et al. (2013) (see p6, L139-142). With careful reading, I think that the data from Olenius et al (2013) is the input data to ACDC, but this should be stated. On p3L87, the table generator is introduced, and is said to require "molecular cluster thermochemistry data for the given chemical compounds, and user-defined input for the ranges of the parameters that define the ambient conditions." You could specify already here that "molecular cluster thermochemistry data for the given chemical compounds" is input to ACDC.

It is correct that both the table generator and the interpolator are applied to obtain the results presented in the Results section, and the cited paper by Olenius et al. (2013) contains the input data for calculating the formation rates. This is now stated on P7, L156-157:

We demonstrate the application and performance of the J-GAIN table generator and interpolator using previously published quantum chemical data…

and on L158 and L161:

Here, the input molecular cluster data…

We apply the $H_2SO_4$–$NH_3$ cluster data to generate tables…

The table generator input is now clarified in Section 2.1 and Appendix A (see the replies to comments 1.1 and 1.2 above). The molecular cluster thermochemistry data is given in ACDC input format, although it can be noted that such data is not relevant to only ACDC, but in practice to any cluster-dynamics-based model with cluster evaporation, as discussed above (see e.g. Yu et al., 2018; Zhang et al., 2012; Li and Signorell, 2021; Elm et al., 2020). The exact input format of the thermochemistry data is now briefly summarized in Appendix A with references to the ACDC manual and example files. The data files can be normally requested from the quantum chemistry researchers who have produced the data.

4. In general, it would be helpful if input data and output data for each part were named a bit more consistently throughout the manuscript. Maybe this could be part of Fig. 1 for example?

We agree, and have modified Figure 1 to show that the generator and the interpolator are two independent parts of the J-GAIN tool, and to indicate which input and output apply for these two parts.

2. Since this tool is meant for implementation into host models, I have looked through the code repository and while it is in general tidy and includes informative readme files, it could be more helpful for users with some small tweaks:

1. At first glance, it is not clear which steps would be needed to run or incorporate J-GAIN in a host model and clarifying this would improve reusability. A step-by-step instruction for use even in the base folder (e.g. which steps would be needed to reproduce some part of the

example case in the manuscript, ideally including all bash commands) would make it clearer what and in which order the different procedures must be performed.

We have now made the following modifications and additions in the GitHub repository:

- Main page:
  - We list brief step-by-step instructions for the whole procedure from table generation to implementation. In order to avoid a longer list of technical details on the main page, we have added detailed instructions and related commands to a new directory that enables reproducing the $H_2SO_4$–$NH_3$ test case (see below).
  - We clarify that the generator and the interpolator are two separate parts of the tool, and that the interpolator is not restricted to tables produced by the J-GAIN generator.
- Subdirectory "GMD_example":
  - The new subdirectory contains detailed step-by-step instructions, including namelist templates and bash commands, for reproducing the $H_2SO_4$–$NH_3$ test case presented in the manuscript. The molecular cluster input data is included as example files in the 'acdc' folder.
  - We also provide an automatic script that generates two tables for the $H_2SO_4$–$NH_3$ example, a coarser table and a reference table, and evaluates the values interpolated from the coarser table by comparison to the exact values in the reference table. The script produces a figure that corresponds to Figure 4 in the manuscript.
  - We clarify that these example files and instructions can be used as a basis for creating and/or applying other tables.
- Other additions/clarifications:
  - Main page and ACDC subdirectory: We also note that the input required for ACDC can generally be requested from quantum chemistry data providers, in case it is not directly available in publications.

2. Additionally, the fortran code does not seem to be commented. I especially think the implementation examples in the example folder (https://github.com/tolenius/J-GAIN/tree/main/examples/interp_dual) would benefit from some commenting to help potential implementers.

We have now added clarifying comments in the main implementation example script dual_table.f90 in the examples/interp_dual/ folder. These comments explain the steps in the implementation (reading in tables, setting input parameters in the call to the interpolation routine, obtaining the interpolated formation rate for the host model), and thus further help the implementer to e.g. add new tables.

We have also added brief comment lines in the various module files in the generator and interpolator folders to clarify the purpose of the subroutines.

Other comments:

1. The software prerequisites in the code repository do not link to anywhere where the software can be installed. The gfortran version needed was not the most recent and the build failed with the most recent version.

We agree that providing links to specific programs is useful, but we feel that it is problematic to provide installation guidance for standard programming languages and/or environments:

- The installation depends on the operating system, and there may also be different approaches for a given system (e.g. sudo apt, homebrew, Cygwin, Windows Subsystem for Linux…)
- Also, programming languages are normally by default included in standard Unix computing systems and in Linux distribution for Windows (e.g. Cygwin) and do not need to be separately installed by the user.
- In general, it is most straight-forward to find up-to-date information by searching documentation for the operating system in question; links to specific approaches may become outdated.

Therefore, we list the relevant programming languages in the main page of the GitHub repository, but feel that discussing options for their installation is beyond the scope of the repository. The ACDC application embedded in the table generator is included in the routines, and does not need to be installed separately.

The most recent gfortran version does in fact work, but we believe that we were able to reproduce the failure: on our system, loading a newer gfortran version when another gfortran module is in use can cause a conflict. This can be solved by uninstalling any conflicting modules.

2. P1,L1-2: " New-particle formation from condensable vapors is a common atmospheric process that has significant but uncertain effects on aerosol particle number concentrations and impacts." Vapors are by definition condensable, so maybe "condensable gasses" is better. Secondly, "impacts" is a bit of a loose end here, consider being more precise on what these impacts could be.

The sentence has been modified to read:

New-particle formation from condensable gases is a common atmospheric process that has significant but uncertain effects on aerosol particle number concentrations and aerosol–cloud–climate interactions.

We have changed "condensable vapors" to "condensable gases" also in the beginning of Introduction.

3. P1,L4: "Such data can be implemented in large-scale climate…" I don't think you can implement "data" in a model, the data can be used to calculate/predict formation rates in the models?

The sentence has been reformulated to read:

Such data can be used in large-scale climate and air quality models through parameterizations or look-up tables.

We also modified another sentence in the Abstract: "Ideally, the implementation of such formation rate data should be easy…" to read:

Ideally, the incorporation of such data should be easy…

4. P1,L5: " Models benchmarked against measurement data provide a straight-forward means to assess formation rates over a wide range of atmospheric conditions for given chemical compounds. " I don't understand what this sentence means in this context.

We agree that the sentence is unclear and have reformulated it as follows:

Molecular cluster dynamics modeling, ideally benchmarked against measurements when available for the given precursor vapors, provides a straight-forward means to calculate high-resolution formation rate data over wide ranges of atmospheric conditions.

5. Fig1: I think this could be a great place to make it clear what parts of the tool does what, and what the user is expected to do. The ACDC input is also not mentioned here as far as I can see.

In revised Figure 1, the input and output of the generator and interpolator parts are now clarified with brief texts and arrows to and from the two boxes that summarize the usage of these parts (see also the replies to main comments 1.3 and 1.4 by reviewer 1). The figure caption has been modified to read:

Flow chart illustrating the generation and application of particle formation rate tables by J-GAIN. The boxes summarize the steps for using the two parts: (1) table generation with automatized calculation of formation rates by cluster dynamics modeling, and (2) implementation of tables and the table interpolator in a host model. User-defined input and output are specified outside the boxes.

6. P4,L94-95: "As the inclusion of charged species and hydrates in a molecular cluster data set requires a significant computational effort, these effects are not always available." It's not clear what "not always available" means here. From the host model?

We agree that the formulation was vague. The challenge in including the ion and hydrate effects is that it requires additional efforts in computing the thermochemistry data set. In order to include ions, the negatively and positively charged counterparts of the electrically neutral clusters must be added in the data set. For including hydration, the water-containing clusters must be added, i.e. those that have otherwise the same molecular compositions as the given set of clusters, but contain also one or more water molecules. This leads to a significant additional workload and computational burden (see e.g. Rasmussen et al., 2020). From the formation rate modeling perspective, however, it is straight-forward to include the ionic species and hydrates once the input thermochemistry is available (Olenius et al., 2013; Henschel et al., 2016).

The sentence has been modified to read:

As the addition of charged species and hydrates in a quantum chemistry data set requires a significant computational effort, these effects are not always included in available thermochemistry data sets.

7. P5,L127: In this section it is not clear to me what the objective is. Are you simply providing some examples of what could be done in a host model? Are the examples related to code you provide? If this section is related to some code you provide, I would make this clear.

Yes, the purpose is to provide an example of a practical application that involves several tables, corresponding to different chemical mechanisms for particle formation. With this, we wish to clarify that the presented look-up table approach is not restricted to a single table, but can be applied to a combination of tables according to the user's needs. While the code repository includes a simple example of additive tables (subdirectory examples/interp_dual, file dual_table.f90), in Section 2.3 we give an example of a possible table combination that the user could construct, depending on which types of particle formation chemistries they wish to include.

The text has been reformulated as follows:

Importantly, the interpolator is not limited to using a single table: separate particle formation pathways, corresponding to different chemical compounds, can be incorporated as separate tables. If more than one table are used, the interpolator is applied separately to each table, and the total formation rate can be obtained as the sum of the individual formation rates.

The repository includes a simple example of summing the rates interpolated from two separate tables. However, the user may construct different ways to treat several tables according to their needs and data availability. To give an example, a possible practical application could be as follows: separate tables are used for parallel formation mechanisms, for example, inorganic $H_2SO_4$–base and organics-driven pathways. There may also be alternative tables that are selected based on the presence of a given chemical species, that is, if the concentration of the species is high enough for the species to

contribute. For instance, there may be data for particle formation from $H_2SO_4$ and $NH_3$ with or without an amine species. In the presence of the amine, a table of $H_2SO_4$–$NH_3$–amine formation rates is selected for the $H_2SO_4$–base pathway, while otherwise a $H_2SO_4$–$NH_3$ table is applied. This example of a potential table combination is schematically presented in Fig. 2.

The caption of Figure 2 has been modified to read:

Schematic presentation of treatment of several tables: an example of a possible table combination that the user may construct according to their needs.

8. P6,l144: "We also generate tables suitable for global applications" it is not clear what "also" means here, is it different to what you present in the sentence before?

This was indeed not clear, and we have rewritten the sentences as:

We apply the $H_2SO_4$–$NH_3$ cluster data to generate tables of different resolution and coverage. First, we demonstrate the effect of table resolution by generating tables that cover subsets of independent parameter ranges where $J$ is sensitive to the parameter values, and compare the interpolated values of $J$ to accurate values given by a high-resolution reference table. Second, we generate extended tables suitable for global applications, where the ranges of all independent parameters cover the various environments from boundary layer to upper troposphere. The extended tables are applied to evaluate the accuracy of $J$ interpolated over the full set of parameters [$H_2SO_4$], [$NH_3$], $T$, CS and IPR that follow representative diurnal cycles, corresponding to practical model implementations.

9. P7,L157-159: I assume the main point here is not the varying over time, but the fact that the interpolation is over all parameters? The way it reads now, it looks a bit like the varying over time should add something particular.

Yes, this is correct. The text has been modified to read:

In order to demonstrate the application of the interpolator for interpolation over all independent parameters at realistic ambient conditions, corresponding to implementation of the routine in an atmospheric model, $J$ is determined for a representative diurnal cycle as shown in Fig. 5.

10. P9,L184: "the run time exhibits a major increase when the table size increases beyond ca. 2^28" Just out of curiosity, why do you think this is?

This appears to be due to memory limitations, which cause slower performance as the array size becomes extremely large and the interpolator needs to retrieve values from different locations of the array. The exact behavior of the run time for very large tables thus depends on the system and its memory management. We have modified the text on P9, L203-206 to read:

In addition, the run time exhibits a major increase when the table size becomes very large, here beyond ca. $2^{28}$ (of the order of $\gtrsim 10^8$) data points. This is due to memory limits and management, and thus the exact threshold size depends on the computing system. For example, for the $H_2SO_4$–$NH_3$ table with $N=5$ and the current simple test set-up with 64 GiB memory on the node, the threshold of $\sim 2^{28}$ points corresponds to $k > 5$.

We also note that the performance can, if needed, be improved by splitting too large tables into subtables (on P9-10, L206-208, and P10, L221-223):

In the case that managing very large tables becomes slow on a given system, the performance can be optimized by splitting the table into subtables that cover different parts of the ambient conditions parameter space.

(…)

If the numbers and ranges of parameters cannot be optimized further, very large tables can be divided into separate subtables that cover different sets of ambient conditions, and selected within the host model application based on the input conditions.

11. P9,L189: Would it not be more effective here to also state how much including the scheme would increase the total run time of the atmospheric component for example?

Yes, aerosol formation rates are essentially part of the atmospheric chemistry description, which is typically among the heaviest components in Earth system and chemical transport models. In the applied EC-Earth3 configuration, atmospheric composition, including gas-phase chemistry and aerosol processes, is described by the TM5 model (van Noije et al., 2014) that is coupled to the atmosphere model IFS (the Integrated Forecasting System of the European Centre for Medium-Range Weather Forecasts). The model run time is dominated by TM5: including TM5 with full chemistry (without the look-up tables) increases the overall run time by up to a factor of ca. 9 (depending on the exact set-up and output; van Noije et al., 2014; personal communication with Carl Svenhag). Therefore, the relative increase in run time upon the inclusion of the table interpolation routine is of the same order for the chemistry component and the full model.

The sentence on P10, L214-216 has been modified to read:

Such contribution can be considered acceptable compared to the overall model run time of $\gtrsim 10$ hours per year, or to the contribution of the atmospheric chemistry component which accounts for up to even ~90 % of the total run time (van Noije et al., 2014).

12. P10,L201: The title reads: "Potential limitations in applying formation rates in a host model", would it not be more precise to say "potential limitations for applying look-up tables in a host model"? There are already parameterisations for formation rates in most models.

In fact, the issues discussed in Section 3.3 apply also to parameterizations, or generally to any pre-calculated or pre-determined formation rate data (whether already included in a model or to be applied through e.g. J-GAIN). We feel that it is useful to bring up these aspects, so that the user understands the possible limitations and uncertainty sources. The limitations listed in the section are relevant regardless of the data format or source: (1) The vapor species (tracers) that the formation rate depends on need to be available in the host model, unless (2) given species can be represented as a lumped compound. (3) In any case, such formation rates always inherently involve the steady-state assumption, because explicit molecular cluster kinetics are not considered (due to the computational burden and complexity in a large-scale framework). Potential issues related to this assumption are extensively discussed in the cited references (e.g. Yu, 2003; Olenius and Roldin, 2022) and summarized on P12, L253-262.

For clarity, we have added the following in the beginning of the section:

It must be noted that incorporating aerosol formation rates in an atmospheric model involves given limitations. These limitations are independent of the data source of implementation method, and apply equally to look-up tables, parameterizations or other approaches.

13. P11, L216-217: please double check this sentence.

The sentence has been rewritten as:

For example, monoamines with similar properties—namely di- and trimethylamines—have been approximated as a single representative alkylamine species, the emissions of which are scaled from ammonia emissions by assumed amine-to-ammonia ratio due to their common sources (Bergman et al., 2015).

**General comments:**

The manuscript by Yazgi et al. presents a tool J-GAIN to generate and interpolate look-up tables of formation rates, allowing the implementation of theoretical particle formation rate data in atmospheric large-scale models. They conducted tests on the application and performance of J-GAIN using theoretical data for $H_2SO_4$-$NH_3$ particle formation, which show that J-GAIN is efficient and accurate. The work is technically well performed and the chosen methodology is appropriate for the purpose of this study. The selected topic should be interesting across a range of atmospheric model development community. The manuscript is well written and easy to follow. Therefore, I recommend publication of this manuscript after a minor revision.

**Specific comments:**

- Line 105, it could be better to present how many points the higher-resolution reference table has. Although it is known that values determined from a table of sufficient resolution are guaranteed to be close to the original data, it is better to give specific relative errors between them in order to strongly convince readers.

    In order to clearer show the numbers of data points along given axes, we have now written the numbers in the form "$2^x + 1 = y$", e.g. "$2^6 + 1 = 65$" in Figures 3–5. The number of points in the highest-resolution table used here is $2^{24} + 1 \approx 2 \times 10^7$.

    It should be noted, though, that the absolute number of points in the reference table is irrelevant since here the table only serves as a source of exact values. The relative errors in interpolated values obtained from the coarser tables are determined at the exact points of the reference table, i.e. by comparison to accurate values. While the examples show errors along the vapor concentration axes, here $[H_2SO_4]$ and $[NH_3]$, the user can perform similar tests also for other independent parameters. We have added example scripts for such tests in the new 'GMD_example' folder in the GitHub repository.

    We have also clarified the principle of the resolution tests, i.e. that the powers of two are used for doubling of resolution, on P7, L172-173:

    That is, for each subsequent table the number of points is doubled, resulting in numbers ranging from 17 to 513.

- Line 211 and line 215, it seems it is inconsistent about whether different amines can be modeled as a lumped compound.

    We agree that this formulation was unclear, and have rewritten the text on P11, L239-241 as follows:

    Different types of amines may exhibit different particle formation efficiencies, but some amine species are similar in terms of their effects on the formation rate (Jen et al., 2014; Olenius et al., 2017).

    Also the sentence on L245-247 has been modified; see the reply to comment 13 by reviewer 1.

- Figure 3(a), it's hard to see any other curves except the curve representing $2^9+1$ points. It would be better to change the color or pattern of these curves, making the figure look more clear.

    It is true that the lines mostly fall on top of each other in Figure 3a, as the differences between them are too small to be distinguished on a logarithmic scale (meaning that the interpolated formation rates are rather accurate even for the coarsest table tested here). The overlap is so significant

that it is difficult to affect the appearance of the figure by line styles, and thus we have now added a note on the overlap in the figure caption:

Note that the different lines mostly fall on top of each other.

The differences between the lines are clearly presented in panels (b) and (c) which show the relative differences in percent. The purpose of panel (a) is to show the behavior of the absolute rate and its slope for understanding the trends in the relative errors (seen in panels (b) and (c)).

We note, though, that we have changed the colors in Figure 3 and other figures to improve the readability considering readers with color vision deficiency.

- Some minor mistakes are shown in the manuscript, e.g., Line58, "be close to the original data" instead of "be close the original data". Please recheck and revise the whole manuscript.

This has been corrected. We have also rechecked the whole text for typos and modified individual words and expressions to improve readability. All changes are marked with yellow highlight.

- Please check the guidelines of Geoscientific Model Development for references, and all the references should be cited in the same style.

We use the Geoscientific Model Development manuscript template, and it appears that some redundant fields in the bibliography file had been automatically included in the reference list. We have now removed these fields and re-checked that the reference styles are correct.

**References**

Elm, J., Kubečka, J., Besel, V., Jääskeläinen, M. J., Halonen, R., Kurtén, T., and Vehkamäki, H.: Modeling the formation and growth of atmospheric molecular clusters: A review, J. Aerosol Sci., 149, 105621, https://doi.org/10.1016/j.jaerosci.2020.105621, 2020.

Henschel, H., Kurtén, T., and Vehkamäki, H.: Computational Study on the Effect of Hydration on New Particle Formation in the Sulfuric Acid/Ammonia and Sulfuric Acid/Dimethylamine Systems, J. Phys. Chem. A, 120, 1886–1896, https://doi.org/10.1021/acs.jpca.5b11366, 2016.

Li, C. and Signorell, R.: Understanding vapor nucleation on the molecular level: A review, J. Aerosol Sci., 153, 105676, https://doi.org/10.1016/j.jaerosci.2020.105676, 2021.

van Noije, T. P. C., Le Sager, P., Segers, A. J., van Velthoven, P. F. J., Krol, M. C., Hazeleger, W., Williams, A. G., and Chambers, S. D.: Simulation of tropospheric chemistry and aerosols with the climate model EC-Earth, Geosci. Model Dev., 7, 2435–2475, https://doi.org/10.5194/gmd-7-2435-2014, 2014.

Olenius, T. and Roldin, P.: Role of gas–molecular cluster–aerosol dynamics in atmospheric new-particle formation, Sci. Rep., 12, 10135, https://doi.org/10.1038/s41598-022-14525-y, 2022.

Olenius, T., Kupiainen-Määttä, O., Ortega, I. K., Kurtén, T., and Vehkamäki, H.: Free energy barrier in the growth of sulfuric acid–ammonia and sulfuric acid–dimethylamine clusters, J. Chem. Phys., 139, 084312, https://doi.org/10.1063/1.4819024, 2013.

Rasmussen, F. R., Kubečka, J., Besel, V., Vehkamäki, H., Mikkelsen, K. V., Bilde, M., and Elm, J.: Hydration of Atmospheric Molecular Clusters III: Procedure for Efficient Free Energy Surface Exploration

of Large Hydrated Clusters, J. Phys. Chem. A, 124, 5253–5261, https://doi.org/10.1021/acs.jpca.0c02932, 2020.

Yu, F.: Nucleation rate of particles in the lower atmosphere: Estimated time needed to reach pseudo-steady state and sensitivity to H2SO4 gas concentration, Geophys. Res. Lett., 30, https://doi.org/10.1029/2003GL017084, 2003.

Yu, F., Nadykto, A. B., Herb, J., Luo, G., Nazarenko, K. M., and Uvarova, L. A.: $H_2SO_4$–$H_2O$–$NH_3$ ternary ion-mediated nucleation (TIMN): kinetic-based model and comparison with CLOUD measurements, Atmospheric Chem. Phys., 18, 17451–17474, https://doi.org/10.5194/acp-18-17451-2018, 2018.

Zhang, R., Khalizov, A., Wang, L., Hu, M., and Xu, W.: Nucleation and Growth of Nanoparticles in the Atmosphere, Chem. Rev., 112, 1957–2011, https://doi.org/10.1021/cr2001756, 2012.

---

## Author Response (AR2)

We thank the editor for the review, and have made the suggested updates.

The revised version of the J-GAIN code repository on GitHub is now included as a frozen copy in the Zenodo repository (zenodo.org/record/8220223), as well as in the form of a new release in the GitHub repository (github.com/tolenius/J-GAIN/releases/tag/v1.1.0).

The revised version with the additional instructions and examples is now numbered v1.1, which we have updated in the manuscript title.